# Temporal Splicing Switches in Elements of the TNF-Pathway Identified by Computational Analysis of Transcriptome Data for Human Cell Lines

**DOI:** 10.3390/ijms20051182

**Published:** 2019-03-08

**Authors:** Nikolai Genov, Alireza Basti, Mónica Abreu, Angela Relógio

**Affiliations:** 1Charité—Universitätsmedizin Berlin, Corporate Member of Freie Universität Berlin, Humboldt—Universität zu Berlin, and Berlin Institute of Health, Institute for Theoretical Biology, 10115 Berlin, Germany; nikolai.genov@zedat.fu-berlin.de (N.G.); alireza.basti@charite.de (A.B.); monica_tpa@live.com.pt (M.A.); 2Charité—Universitätsmedizin Berlin, Corporate Member of Freie Universität Berlin, Humboldt—Universität zu Berlin, and Berlin Institute of Health, Medical Department of Hematology, Oncology, and Tumor Immunology, Molecular Cancer Research Center, 13353 Berlin, Germany

**Keywords:** circadian clock, cancer, systems biology, high-throughput data analysis, alternative splicing, analysis of oscillating genes, pathway enrichment, RNA-sequencing, TNF-pathway

## Abstract

Alternative splicing plays an important role in numerous cellular processes and aberrant splice decisions are associated with cancer. Although some studies point to a regulation of alternative splicing and its effector mechanisms in a time-dependent manner, the extent and consequences of such a regulation remains poorly understood. In the present work, we investigated the time-dependent production of isoforms in two Hodgkin lymphoma cell lines of different progression stages (HD-MY-Z, stage IIIb and L-1236, stage IV) compared to a B lymphoblastoid cell line (LCL-HO) with a focus on tumour necrosis factor (TNF) pathway-related elements. For this, we used newly generated time-course RNA-sequencing data from the mentioned cell lines and applied a computational pipeline to identify genes with isoform-switching behaviour in time. We analysed the temporal profiles of the identified events and evaluated in detail the potential functional implications of alterations in isoform expression for the selected top-switching genes. Our data indicate that elements within the TNF pathway undergo a time-dependent variation in isoform production with a putative impact on cell migration, proliferation and apoptosis. These include the genes *TRAF1*, *TNFRSF12A* and *NFKB2*. Our results point to a role of temporal alternative splicing in isoform production, which may alter the outcome of the TNF pathway and impact on tumorigenesis.

## 1. Introduction

Alternative splicing (AS) belongs to one of the key biological processes, which regulates gene expression diversity and function in eukaryotes. In this process, different coding regions of the pre-mRNA are included in the mature mRNA in a combinatorial way. This leads to exon/intron inclusion or exclusion events in the final transcript form of a gene. As a result, AS is able to generate a myriad of functional proteins from the same gene. This affects cellular functions and pathways, and provides the organism an evolutionary advantage to adapt to its changing environment. Remarkably, it was estimated that around 95% of human genes are alternatively spliced [1,2]. Moreover, AS can influence different aspects of protein functionality including protein localization, protein–protein binding and catalytic properties of proteins [3]. Also cellular phenotypes (e.g., apoptosis and proliferation), as well as regulation of gene expression and histone modification can be fine-tuned by AS events [3]. Therefore, failures in splicing decisions (e.g., due to genetic mutations leading to sequence alterations within the transcript, which may alter the sequences of splice sites) or within the splicing machinery (e.g., in small nuclear ribonucleic particles snRNPs) may lead to the generation of aberrant transcripts and contribute to human disease [4,5].

In cancer, changes of AS are well studied, and a deregulated AS is reported as one of the hallmarks of cancer [6,7]. Several protein isoforms generated by alternative splicing are crucial for cancer progression and are the subject of experimental therapeutic interventions [8]. One example is the RON (tyrosine kinase receptor) gene, which is involved in cell motility and invasion. A constitutively active isoform of this gene (delta-Ron) is generated via the skipping of an alternative exon regulated by the splicing factors SF2/ASF. It was shown that overexpression of SF2/ASF isoform translates into the constitutively active form of the RON kinase that determines the epithelial to mesenchymal transition facilitating invasion, intravasation and extravasation, known steps of metastasis [9]. Another example is the splicing factor SRSF6, which via a fine-tuning of its expression, is able to alter the splicing of several oncogenes, including the insulin receptor (INSR), the kinase Mnk2 (MKNK2) and the discs large MAGUK scaffold protein 1 (DLG1) leading to the production of oncogenic isoforms or to the reduction of the tumour-suppressing isoforms [10]. 

Several studies have illustrated a cancer-specific pattern of AS in various cancer types (e.g., in neuroblastoma, Hodgkin lymphoma (HL), prostate, colon and lung cancers) [11,12,13,14,15]. In particular, the analysis of AS using microarray data in HL revealed stage–specific patterns of AS between HL cell lines of different tumour stages [13]. Interestingly, HL tumours of the same stage (HDML-2 and L-540, stage IV) showed greater similarity in their splicing pattern (70% similarity) when compared to a HL cell line from a different stage (HD-MY-Z, stage IIIb, 55% similarity). These patterns also enabled the establishment of a differential genetic signature between HL tumours and a non-tumour B lymphoblastoid cell line (LCL-HO). Additionally, several splicing factors (e.g., hnRNP E1, U2AF65 and NOVA-1/2) were reported to be differentially expressed in the HL cell lines as compared to the B lymphoblastoid cell line (LCL-HO). These findings point to the relevance of splicing in tumorigenesis for this cellular model of HL [13]. 

Increasing evidence points to the regulation of several post-transcriptional processes in a time- dependent manner. Genes and proteins hallmarked by circadian (ca. 24 h period) and ultradian (between 8–12 h period) expression were shown to be conserved across several species and regulate several pathways [16]. These include AS, polyadenylation, mRNA stability and mRNA transport [17]. Additionally, several RNA-binding proteins involved in AS regulate the expression, localization and translation of core-clock components (e.g., *Per1* and *CLOCK*) [18,19,20]. Altogether, these findings indicate a role of splicing in the regulation or fine-tuning of the circadian machinery and vice versa with a potential impact in tumorigenesis. 

One of the pathways for which published data points to a role of AS in the regulation of elements of the signalling cascade is the TNF (tumour necrosis factor) pathway [21]. As demonstrated by many studies, TNF plays a central role in diverse cellular events such as inflammation, immunity, cell proliferation, differentiation and apoptosis [22,23,24]. Hence, TNF can activate several downstream elements to induce differential cellular responses through the activation of TNF receptor elements (TNFRs), members of the TNF superfamily receptors. This activation, in turn, can induce the initiation of diverse signalling cascades leading to apoptosis (via the TNFR1-associated death domain protein TRADD) or inflammation and proliferation (via the activation of TNF receptor-associated factors TRAFs, NF-kB activation as well as the c-Jun N-terminal kinase (JNK)-dependent kinase cascade) [25]. Consequently, TNF is involved in many aspects of tumorigenesis by inducing tumour suppression (activating apoptosis as described above), as well as tumour initiation and progression. This is achieved mostly via the activation of NF-kB- and activator protein 1 (AP-1)-dependent pathways [24,25], or enhancing angiogenesis through various angiogenic factors such as vascular endothelial growth factor (VEGF) and the JNK-dependent pathway [26]. Given the AS properties described above, it is conceivable that AS-generated isoforms within the elements of the TNF signalling pathway may play a major role in the balance between TNF-induced survival, inflammation and apoptosis. 

Despite the increasing relevance of the time-dependent regulation of cellular processes in cancer progression, the impact of circadian regulation in splicing remains poorly understood. Thus, it is of great biological and clinical relevance to identify timely regulated AS events among elements of the TNF pathway in cancer cellular models. Yet, additional studies at the transcriptome level using circadian data sets are still rare and are essential for the investigation of time-dependent AS regulation [27,28]. 

In this work, we characterized the extent of a time-dependent isoform generation in elements within the TNF pathway. We generated 33 time-course data sets (sampling every three hours between ZT12 h and ZT42 h) to investigate the time-dependent production of isoforms in two HL cell lines at different stages (HD-MY-Z, stage IIIb and L-1236, stage IV) compared to a B lymphoblastoid cell line (LCL-HO) with a focus on TNF pathway-related elements. ZT (zeitgeber time) is defined as the time (in hours) after cell synchronization with medium change. Using a computational approach, we identified several genes involved in the TNF pathway to undergo a time-dependent change in isoform expression with potential implications on the output of the pathway. These include TNF receptor-associated factor 1 (TRAF1), TNF receptor superfamily 12A (TNFRSF12A) and the nuclear factor kappa B subunit 2 (NFKB2) with relevant functions in cell proliferation, migration and apoptosis. Our results further emphasize the potential impact of AS in cellular physiology, in a cancer context, and highlight the effect of temporal isoform switching events as mediators of these processes.

## 2. Results

### 2.1. Computational Analysis of RNA-Seq Time Series Data Reveals a High Number of Time-Dependent Splicing Switches between Isoforms in the LCL-HO, HD-MY-Z and L-1236 Cell Lines

Previous research pointed at the relevance of splicing in cancer, and identified cell-specific AS changes in HL cell lines [4,13,29]. In order to investigate the time-distribution of the isoforms detected in haematological cell lines we used a B lymphoblastoid cell line LCL-HO, the Hodgkin lymphoma cell lines HD-MY-Z (stage IIIb) and L-1236 (stage IV) and generated a total of 33 time-course data sets (sampling every three hours between ZT12 h and ZT42 h), which were subsequently sequenced. We used the 33 data sets as the input for our computational analysis pipeline (Figure 1A). We quantified paired-end RNA-Seq data with the Salmon software by using the European Molecular Biology Laboratory (EMBL) reference transcriptome Homo_sapiens.GRCh38.cdna. We used the quantified transcripts in TPM (transcripts per million) to analyse the isoform expression profiles over time, identified switches in the expression of isoforms from the same set of genes with the TSIS R package [30], and further investigated the switches regarding their potential functionality. Isoform switching is defined as the reversal of relative abundance of different isoforms of the same gene over time. Interestingly, the number of switches and their distribution over time varies greatly between the cell lines (Figure 1B). We filtered the switching events with a *p*-value cut-off of 0.001, and a difference cut-off of 1 as suggested in the TSIS package (see Materials and Methods for details) [30]. The difference cut-off is defined as the sum of the average abundance differences in TPM of the two isoforms in the intervals before and after the switch. This indicates the magnitude of the switch. Higher values mean larger changes in expression before and after the switch. Once we identified potential switches, we performed a downstream analysis with the IsoformSwitchAnalyzeR R package [31] for coding potential and signatures of functional domains. Furthermore, we performed a Reactome pathway enrichment analysis for the groups of genes containing isoform switches. For the LCL-HO, we detected 881 switches in pairwise isoform expression in time (Appendix A) with one switch time peak (LCL-HO_STP1) containing 273 switching events in pairwise isoform expression in a 3 h bin between 27 h and 30 h. A STP is defined as a peak in the distribution of isoform switching events per 3 h bin as computed by the TSIS algorithm. For the HD-MY-Z we detected 1865 switches between isoforms (Appendix A) and one STP (HD-MY-Z_STP1) with 783 switching events in pairwise isoform expression in a 3 h bin between 33 h and 36 h. For the L-1236 cell line we detected 755 switches between isoforms (Appendix A) and three STPs at 18 h to 21 h (L-1236_STP1, 166 switching events), 24 h to 27 h (L-1236_STP2, 167 switching events) and 36 h to 39 h (L-1236_STP3, 148 switching events), as depicted in Figure 1B. Interestingly, this might indicate a specific pattern of STP distribution that could correlate to the clinical stage of the cancer (HD-MY-Z vs L-1236), so that the higher stage cell line contains more STPs. 

### 2.2. The Switch Time Peaks (STPs) Are Characterized by Trends in Functional Changes such as the Transcript Coding Potential, Domain Contents and Sensitivity to Nonsense-Mediated Decay

We further investigated the switches between isoforms in time, focusing in the region around the STPs, and tested for potential functional properties of the resulting transcripts at the sequence level (Figure 2, Appendix A). Such properties include the coding potential of the transcripts (CP), the potential protein domains identified based on the sequence of the transcript (domains identified), intron retention properties (IR), the susceptibility of the transcript to nonsense-mediated decay (NMD), and the similarity of the ORF sequence between the time-points (ORF seq similarity). We found large differences in the functional outcome of the switching events between the cell lines LCL-HO and HD-MY-Z (Figure 2A,B, respectively). For the LCL-HO cell line the STP between 27 h and 30 h shows an increase of coding transcripts combined with longer ORFs and domain gains (Figure 2A), however this difference is not statistically significant (Appendix A). For the STP of the HD-MY-Z cell line between 33 h and 36 h we detected a significant increase (Appendix A) of the number of non-coding transcripts and, therefore, domain loss in the sequence with corresponding shorter ORFs (Figure 2B). The three STPs of the L-1236 cell line lead to interesting observations (Figure 2C–E). The STP1 between 18 h and 21 h shows a statistically significant (with confidence interval of 95%) increase of coding transcripts with domain gains and longer ORFs (Figure 2C and Appendix A). The second and third STPs (STP2, between 24 h to 27 h and STP3, 36 h to 39 h, respectively) show several switches towards isoforms with shorter ORFs and domain loss (Figure 2D,E, respectively). Based on our findings, the STPs between 24 h to 27 h and 36 h to 39 h in the L-1236 cell line are similar to the STP of HD-MY-Z between 33 h and 36 h, which could indicate a potential relevance of the temporal distribution of the STPs to its functional outcome in each cell line. The changes in transcript length and potential functionality between the STPs relate to changes in the alternative splicing events over time (Appendix A). In all cases, alternative transcription stop- and start-sites are the mechanisms by which most of the changes in isoform balance are generated. 

### 2.3. The Genes for which Switches in Isoform Expression Were Detected in Each Cell Line Show a Specific Enrichment of Functionality Focused on the SLIT/ROBO, Sterol Regulatory Element-Binding Proteins (SREBP) and Circadian Clock Pathways

The switches detected for all cell lines, at the gene level, vary between 547 genes with switches between isoform-pairs for the LCL-HO cell line (Figure 3A, Appendix A), 958 genes with switches for the HD-MY-Z cell line (Appendix A), and 460 genes with switches for the L-1236 cell line (Appendix A). Some genes have multiple pairs of isoforms switching in time hence the number of switching events is greater than the number of genes. A core set of 89 genes with isoform switches is common between the three cell lines (Appendix A). This core set shows enrichment of Reactome pathway terms related to translation and splicing (adjusted (BH) *p*-value = 8.360 × 10^−7^) as well as NMD (adjusted (BH) *p*-value = 1.870 × 10^−3^) (Figure 3B, Appendix A). Additionally, we performed an enrichment analysis for disease ontology pathways using the 89 gene sets (Figure 3C, Appendix A). The genes with switches identified by our data (e.g., *VEGFA*) are involved in a variety of cancers including skin, pancreatic and colon cancer, as well as lung disease according to our enrichment analysis, which is in agreement with previous studies for the mentioned gene [32]. However, when analysing the gene sets of the individual cell lines for which switching in isoform production was detected we observed a more diverse scenario (Figure 3D, Appendix A). For the B lymphoblastoid LCL-HO cell line (547 genes with identified switches), there is an enrichment of ROBO-related pathways (Appendix A, adjusted (BH) *p*-value = 7.099 × 10^−8^). The SLIT/ROBO pathway has important roles in tumorigenesis, cancer progression and metastasis and thus acts as a regulator for multiple oncogenic signalling pathways including the CD20, mTOR (mammalian target of rapamycin), VEGF and EGFR (epidermal growth factor receptors) pathways [33]. The ROBO functionality can also be found in the enriched pathways for the HD-MY-Z cell line (Appendix A) regulation of expression of SLITs and ROBOs, (adjusted (BH) *p*-value = 6.918 × 10^−3^), interestingly only in the genes with switches that are unique to this cell line (620 unique genes with identified switches, Appendix A), hence different genes of the same pathway are affected. The L-1236 cell line (460 genes with identified switches), however, shows an interesting enrichment of the FGFR2 (Appendix A, fibroblast growth factor receptor 2, adjusted (BH) *p*-value = 3.640 × 10^−4^) signalling pathway, which is associated with disrupted organogenesis leading to cancer [34]. Additionally, we performed an enrichment analysis on the set of unique genes for each cell lines for Reactome pathways and disease ontology (Appendix A). In addition to RNA splicing and translation-related pathways, we found metabolic (metabolism of steroids, *p*-value = 4.542 × 10^−2^), the SREBP (sterol regulatory element-binding proteins) pathway (regulation of cholesterol biosynthesis by SREBP, *p*-value = 5.090 × 10^−3^ and activation of gene expression by SREBF, *p*-value = 4.542 × 10^−2^) and the circadian clock pathway (*p*-value = 4.542 × 10^−2^) to be enriched in the L-1236 cell line (Appendix A). SREBP belongs to transcription factors that regulate the expression of genes required for the synthesis of fatty acids and cholesterol and thus is relevant for cellular metabolism. For the LCL-HO cell line, the disease ontology enrichment for the unique switching genes shows cancer-related terms (Appendix A). For the HD-MY-Z cell line, this result is more striking with 7 out of the top 10 disease ontology-enriched terms being cancer-related (Appendix A).

### 2.4. Several Genes Display Switching in Isoform Production Over Time

We computed the difference in expression between the switching isoforms with the TSIS package [30] and generated lists with all switching events for the three cell lines (Appendix A). We plotted several gene isoforms out of the top 20 events (Figure 4) to illustrate the temporal changes in the expression of the isoforms. For the LCL-HO (Figure 4A) cell line *HNRNPH1* (heterogeneous nuclear ribonucleoprotein H1) shows an absolute difference of expression of 701.44 TPM between the isoforms ENST00000262584.7 and ENST00000526668.5. *JCHAIN* (joining chain of multimeric IgA and IgM) shows an absolute difference in expression of 107.69 TPM between the isoforms ENST00000510437.5 and ENST00000543780.5 and *NME2* (NME/NM23 nucleoside diphosphate kinase 2) shows an absolute difference in expression of 408.09 TPM between the isoforms ENST00000513177.5 and ENST00000512737.5. The *HNRNPH1* gene is involved in splicing. It is overexpressed in many cancer types (e.g., rhabdomyosarcoma, esophageal squamous cell carcinoma and breast cancer) and is required for cancer growth and development [35,36,37]. Notably, the expression of the two isoforms of *HNRNPH1* is different over time for the three investigated cell lines (Appendix A). There is no switch between the expression of the same two isoforms of *HNRNPH1* in HD-MY-Z and multiple switches in our data for L-1236. JCHAIN is involved in innate and adaptive immune response, and is differentially expressed in different cancer types (e.g., colon and prostate cancer) [38]. NME2 is involved in myeloid differentiation, apoptosis and cell development. It acts as a transcriptional activator of the *MYC* gene and its isoforms, which are either differentially or ubiquitously expressed [39]. 

For the HD-MY-Z (Figure 4B) cell line the *CD44* gene shows an absolute difference in expression between the isoforms ENST00000263398.10 and ENST00000352818.8 of 164.95 TPM. The *TNFRSF12A* gene shows an absolute difference in expression of 211.08 TPM for the isoforms ENST00000326577.8 and ENST00000341627.5 and the *HNRNPA1* (heterogeneous nuclear ribonucleoprotein A1) gene (involved in splicing and mRNA transport) shows an absolute difference in expression of 171 TPM for the isoforms ENST00000340913.10 and ENST00000547566.5. *CD44* encodes for a cell-surface glycoprotein and can act both as a growth- and invasiveness-promoting molecule, and as a tumour-suppressing cofactor via its different splice variants [40]. TNFRSF12A belongs to the TNF superfamily and is involved in angiogenesis, cell migration and apoptosis. 

For the L-1236 cell line (Figure 4C), the *NFKB2* gene shows an absolute difference of 84.39 TPM between the isoforms ENST00000189444.10 and ENST00000467116.5, the *PPP4C* (protein phosphatase 4 catalytic subunit) gene shows an absolute difference of 147.32 TPM between the isoforms ENST00000279387.11 and ENST00000561610.1. The *HSPD1* (heat shock protein family D (Hsp60) member 1) gene shows an absolute difference in expression of 348.472 TPM between the isoforms ENST00000388968.7 and ENST00000345042.6. *PPP4C* mediates the repair process of damaged DNA upon double strand break and *HSPD1* is involved in B and T cell activation, among other processes. NFKB2 is a member of NF-KB family of proteins and of the TNF signalling pathway. It is involved in transcription, differentiation, apoptosis and inflammatory response. Altogether, these results indicate that a time-dependent change in isoform expression is present in genes with an impact on vital cellular processes. 

### 2.5. Multiple Genes Involved in the Tumour Necrosis Factor (TNF) Pathway or Related to TNF Signalling Switch between Isoforms in a Time-Dependent Manner

Interestingly, several of the top switching genes such as *TNFRSF12A* and *NFKB2* are related to the TNF pathway. Therefore, we investigated the TNF pathway in detail and detected multiple switches in isoform expression over time in several genes. These included *TRAF1* and *RACK1* (receptor for activated C kinase 1) in the LCL-HO cell line (Figure 5A), *RACK1* and *SHARPIN* (SHANK associated RH domain interactor) in the HD-MY-Z cell line (Figure 5B), as well as in *UBB* (ubiquitin B) and *RPS27A* (ribosomal protein S27a) for the L-1236 cell line (Figure 5C). 

The presence of switches in isoform expression over time and their concentration at specific time points has potential functional implications on the function of the generated protein and the subsequent alterations in the phenotype of the cell. To investigate this aspect in more detail, we performed an analysis at the sequence level with the R package IsoformSwitchAnalyzeR that allows for the integration of data from tools such as CPAT (coding-potential assessment tool) [41] for coding potential analysis and PFAM (protein family) [42] for potential protein domain identification. We focused on the genes *TNFRSF12A, CD44* and *NFKB2* in more detail given their role in the multiple cellular pathways and cancer and the high scoring of the corresponding switching events. We investigated all isoforms with high expression for the corresponding genes. We observed large changes in expression for the *TNFRSF12A* gene in the HD-MY-Z cell line (Figure 6A). For *TNFRSF12A* the switch between the isoforms ENST00000326577.8 and ENST00000341627.5 leads to a change between a coding and a non-coding isoform, meaning that the corresponding stn_TNFRSF12A domain will be affected and the non-coding isoform contributes more to the summed up expression of the gene. Similarly, while multiple isoforms of *CD44* are affected by changes over time, this effect is most dramatic regarding the coding isoform ENST00000528086.5, with a signal that nearly disappears within 9 h (Figure 6B).The overall cumulative expression of the gene shows only a minor change of 4.05% from 773.27 TPM to 804.55 TPM, underscoring the importance of isoform-level investigation. Similarly, the genes *NME2* and *JCHAIN* in LCL-HO show larger changes between the expression of isoforms than the total expression of the gene (Appendix A). For the L-1236 cell line, we identified a switch with potential major implication between the isoforms ENST00000189444.10 and ENST00000467116.5 of *NFKB2*. ENST00000467116.5 does not contain the important death domain, central in the regulation of apoptosis and inflammation. Additionally the multiple Ank (Ankyrin) domains are absent. These are critical for folding and stability and are involved in protein-protein interactions. As mentioned above, the cumulative expression of the gene shows little variations (6.93% from 411.96 TPM to 382.52 TPM) whereas the expression of the isoforms shows strong differences within six hours. These findings point to the potential implication of temporal switching events in isoform expression and its impact in the subsequent protein functions.

The abundance of time-dependent changes of isoform expression in the TNF pathway (Figure 7) points to a possible change of the output of the pathway within a day. The TNF pathway is activated through the binding of TNF to its receptors (TNFR1 and TNFR2) leading to the activation of several signalling cascades. While the activation of the TNFR1 mainly results in the activation of cellular components leading to cell apoptosis, TNFR2 induction is required for cell survival and inflammation responses [25]. One of the main components of the latter pathway is the TRAF complex (TRAF1, TRAF2 and inhibitor of apoptosis proteins cIAPs), which regulates several downstream signalling pathways, including survival and inflammation through the NF-kB pathway and cell apoptosis via interactions with the LUBAC (linear ubiquitin assembly complex) [24,25]. 

Results from our computational analysis revealed that in all of the investigated cell lines, we found a TNF or TNF-related element with temporal changes in isoform production. We identified *TRAF1* in the LCL-HO cell line, *TNFRSF12A* and *RACK1* in the HD-MY-Z cell line and *NFKB2* in the L-1236 cell line, which show significant isoform switching behaviour and can regulate the resulting output of the TNF signalling pathway in this cellular model system.

## 3. Discussion

In the present study, we characterized the existence of time-dependent AS events in HL cell lines (HD-MY-Z, stage IIIb and L-1236, stage IV) and a non-tumour B lymphoblastoid cell line (LCL-HO) using a computational approach. Stage-specific patterns of AS and splicing factor expression for HL cells from different tumour stages has previously been reported [13]. The authors made use of these patterns to classify the stage of each tumour and to distinguish between a non-tumour B lymphoblastoid cell line and cancer cells [13]. To date, no correlation between the splicing events and the circadian clock have been investigated for this cellular system.

### 3.1. Differential Number of Splicing Switches in Cancer Cell Lines Correlate with Tumour Progression Stage

Published data points to the regulation of AS, and various post-translational processes by the circadian clock pointing to a time-dependency of such molecular mechanisms with implications in cellular function [17,27]. To investigate this in detail, we performed a time-course genome-wide screening of the two HL cell lines and the non-tumour B lymphoblastoid cell line and carried out a detailed computational analysis of the generated 33 sets of RNA-seq data. We identified cell-specific isoform switching events and predicted the functional outcome of the identified switches. Our data indicates a distinct and unique pattern of switching behaviour in the investigated cell lines, which could possibly correlate with their specific tumour stage. Comparing the two HL cell lines in our study, we identified a higher number of STPs (three) available in the higher stage cancer cell line (L-1236, stage IV) than the lower stage cell line (HD-MY-Z, stage IIIb) with only one STP (Figure 1B). Moreover, while changes in the potential consequences (such as ORF length and domains present) in the STP of LCL-HO are present, these are not statistically significant. In contrast, in HD-MY-Z, which displays one STP, the changes in the potential functional consequences of pairwise expression switching are significant. Also, for L-1236, the STPs have significant changes and even opposing changes in the potential consequences of the pairwise expression switching of isoforms. Through enrichment analysis for the genes that undergo temporal changes in isoform expression in each cell line, we identified several pathways, which are related to cancer (SLIT/ROBO and FGFR2 signalling pathway). Previous studies indicate that the SLIT/ROBO pathway acts as master regulator for multiple oncogenic signalling pathways including the CD20, mTOR, VEGF and EGFR pathways in several cancer types including lung cancer, colon cancer and lymphoma [33]. Similar to the SLIT/ROBO pathway, the FGFR2 signalling pathway plays a role in cancer progression as well. Since FGFR2 pathway interacts with major signalling pathways, including BMP, WNT, Notch and Hedgehog, its aberrant activity is associated with developmental defects resulting in metabolic disorders, and cancer including lymphoma (reviewed in [34]).

We further analysed a reduced set of genes that present splicing switches in a time-dependent manner. In particular, out of the top 20 genes with largest expression difference between the isoforms we identified the *JCHAIN*, *NME2* and *HNRNPH1* genes (for the LCL-HO cell line), *CD44*, *TNFRSF12A* and *HNRNPA1* (for the HD-MY-Z cell line) and *NFKB2*, *PPP4C* and *HSPD1* (for the L-1236 cell line), which have an impact in tumorigenesis. Transcript variants of *HNRNPH1* were shown to be differentially involved in tumorigenesis. Overall, positive expression of HNRNPH1 is associated with poor tumour differentiation degree [35,36]. The *NME2* gene can act as a metastasis suppressor by altering telomere function and telomere length in lung cancer and fibroblasts [39,43]. CD44 is epigenetically regulated in Hodgkin and non-Hodgkin lymphomas and is reported to be unmethylated in the L-1236 cell line [44]. Several studies have identified various isoforms of the *CD44* gene [45]. Particularly, the CD44v isoforms are involved in specific signalling pathways. While the CD44v3 isoform bind several heparan sulfate-binding growth factors such as fibroblast growth factors (FGFs) and heparin-binding epidermal growth factor (EGF), the CD44v6 isoform contains a binding site for hepatocyte growth factor (HGF) and VEGF inducing CD44v6/HGF/cMet complex formation, leading to c-Met or HGF-induced Ras signalling activation [45]. Cancer cells often express a variety of CD44 isoforms, which enables the regulation of multiple oncogenic pathways required for cancer progression and tumour development [40,46,47]. The *PPP4C* gene is required for the correct DNA damage repair functioning upon a double strand break. Moreover, it was shown to be involved in the TNF and NF-kB pathways. It acts as a signalling component connecting the TNF signal to the JNK pathway [48]. *HSPD1* encodes a heat-shock family protein chaperone that exhibits elevated expression in Hodgkin’s and large cell lymphoma and is involved in the B and T cell immune response [49]. 

Altogether, these genes encode for functionally different isoforms involved in vital cellular and tumorigenic processes and hence, the temporal change in their isoform balance could imply subsequent changes in tumour progression. 

### 3.2. Putative Impact of Splicing Switches in the Fine-Tuning of the TNF Pathway: Survival *vs.* Apoptosis

It was recently reported, that splicing isoforms of the main elements within the TNF pathway can have an impact on both inflammation and cancer and, hence, affect the outcome of the pathway [21]. Among the genes identified by our bioinformatics approach, are several members of the TNF and TNF-related signalling pathway that show a rhythmic pattern of isoform expression: *TNFRSF12A* and *SHARPIN* in the HD-MY-Z cell line, *NFKB2* in the L-1236 cell line and *TRAF1* in the LCL-HO cell line. The TNF receptor superfamily 12A gene (*TNFRSF12A*) displays a time-dependent isoform generation in the HD-MY-Z cell line. Surprisingly, one of the transcripts for which a temporal isoform generation was detected undergoes NMD, which may influence its functional outcome in time. In cancer, TNFRSF12A can promote angiogenesis and endothelial cell proliferation. It was reported that *TNFRSF12A* knockout inhibits hepatocellular carcinoma cell proliferation and migration in vitro [50,51]. TRAF1 belongs to the TNF receptor-associated factors that regulates the activation of NF-kB and JNK pathways and is involved in both cell survival and apoptosis [52,53]. Another important component of the TNF signalling pathway is SHARPIN, which belongs to the LUBAC protein complex involved in the NF-kB and apoptotic signalling pathways. It was shown that SHARPIN deficiency leads to rapid cell death upon TNF stimulation via FADD- and Caspase-8-dependent pathways [54]. Another example of a TNF-related component found in our analysis is the nuclear factor kappa B subunit 2 (NFKB2) for which we identified splicing switches in the L-1236 cell line. Studies have indicated that NFKB2 may be involved in HL development (in L-1236) and the NF-κB-mediated transcriptional repression of the clock feedback limb could cause circadian disruption in response to inflammation in mice [55,56]. In the LCL-HO cell line, TNF receptor associated factor 1 (*TRAF1*) showed a temporal pattern of isoform expression. Interestingly, yeast two-hybrid interaction assays revealed that TNFRSF12A binds to members of the TRAF family including TRAF1 and thereby activates the NF-κB signalling pathway [35]. Components of the NF-κB signalling pathway are affected by AS events, which could either inhibit or enhance the outcome response [57]. 

Taken together, the results of our study further reinforce the hypothesis, that AS could potentially fine-tune the outcome of the TNF signalling pathway in a time-dependent manner. This adds a hidden layer of temporal regulation in the TNF pathway that will require further experimental studies to fully validate and further evaluate the impact of our findings in tumorigenesis. 

## 4. Materials and Methods

### 4.1. Cell Culture

The Hodgkin lymphoma cell lines, HD-MY-Z (DSMZ—#ACC 346) and L-1236 (DSMZ—#ACC 530), and the B lymphoblastoid cell line, LCL-HO (DSMZ—#ACC 185) were maintained in RPMI 1640 (Gibco) supplemented with 1% penicillin-streptomycin (Gibco) and 10% foetal bovine serum (Gibco) and incubated at 37 °C with 5% CO_2_ atmosphere.

### 4.2. Sampling and RNA Extraction

Cells were seeded in 35 mm dishes with a density of 5 × 10^5^ cells/dish and synchronized by fresh medium addition. Cells were collected every three hours, in triplicates and the pellets were stored at −80 °C. Total RNA was isolated using the RNeasy extraction kit (Qiagen), including DNase digestion, according to manufacturer’s instructions. Cells were lysed with 350 µL RLT buffer and the lysate was homogenized. Total RNA concentration was determined using NanoDrop^®^ ND-1000 ultraviolet–visible (UV–Vis) spectrophotometer and stored at −80 °C. A total of 33 time-course data sets (sampling every three hours between ZT12h and ZT42h) was produced for subsequent sequencing.

### 4.3. RNA-Sequencing

For RNA-sequencing the total RNA was processed by Illumina standard protocols to prepare the mRNA-Seq library, followed by deep sequencing in a bi-directional, 75-base mode on an Illumina NextSeq sequencer. The RNA-sequencing was performed in Genomics Core Facility at the European Molecular Biology Laboratory (EMBL) in Heidelberg.

### 4.4. Enrichment Analysis

Enrichment analysis was performed with the DOSE [58] (ver. 3.8.2) for Disease Ontology and ReactomePA (ver. 1.26.0) R packages using Reactome Pathways. All reactome pathway enrichment was performed with a *p*-value cut-off of 0.05 and Bonferroni Hochberg correction for multiple testing against default “universe” background set of genes with a q-value cut-off of 0.2. Disease ontology enrichment was performed with the disease ontology as ontology of choice and *p*-value cut-off of 0.05 as well as q-value cut-off of 0.5. In all cases, the top 10 enriched terms (if present) were plotted.

### 4.5. RNA-Seq Data Processing

The time course paired-end RNA-seq samples were quantified with the Salmon software package against the reference transcriptome provided by the EMBL. The transcriptome version used was Homo_sapiens.GRCh38.cdna.A single sample represents each time point, no replicates were used. Noise was added to the data to simulate potential biological variation with the R function jitter and “factor” parameter set to 10 as well as amount parameter set to 2. 

### 4.6. Detection and Visualization of Splicing Switches

For the detection of splicing switches over time the TSIS R package (ver. 0.2.0) was used with “mean” as the detection method and a spline degree of 4 to focus on isoforms with a period close to 24 h. Isoform switches were visualized with the TSIS [30] package and in detail with the IsoformSwitchAnalyzeR R package. For the TSIS package, the MEAN scoring method was used. Filtering of the results was done with TSIS with a probability cut-off of 0.5, difference cut-off of 1, *p*-value cut-off of 0.001, a minimal time in interval of 2, correlation cut-off of 0, lower time of 0 h and upper time of 30 h (experimental time) without a reduction to the most abundant isoforms. For the analysis of coding functionality, we used a coding potential cut-off of 0.725. For the visualization of pairwise comparisons between the STPs we used a difference cut-off of 0.1. Genes with single isoform are removed from the pairwise STP analysis. 

Computation of abundance differences (from Guo et al. [30]): Metric 2 indicates the magnitude of the switch. Higher values mean larger changes in abundances before and after the switch. S2 is the sum of average abundance differences of the two isoforms in both intervals: *S*_2_(*iso_i_*, *iso_j_*|*I*_1_, *I*_2_) = *d*(*iso_i_*, *iso_j_*|*I*_1_) + *d*(*iso_i_*, *iso_j_*|*I*_2_)(1)
where *d*(*iso_i_*, *iso_j_*|*I_k_*) is the average difference of abundances between *iso_i_* and *iso_j_* in the interval *I_k_*, *k = 1,2* is defined as: *d*(*iso_i_*, *iso_j_*|*I_k_*) = (*1*/|*I_k_*|)∑*_mIk_*|*exp*(*iso_i_*|*s_mI_k__*, *I_k_*) − *exp* (*iso_j_*|*s_mIk_*, *I_k_*)|(2)
|*I_k_*| is the number of samples in interval *I_k_* and *exp*(*isoi*|*s_mIk_*, *I_k_*) is the expression of *iso_i_* of sample *s_mIk_* in interval *I_k_*. 

### 4.7. Data Access

The RNA-sequencing data was submitted to the GEO repository and will be released upon publication of the manuscript.

## Figures and Tables

**Figure 1 ijms-20-01182-f001:**
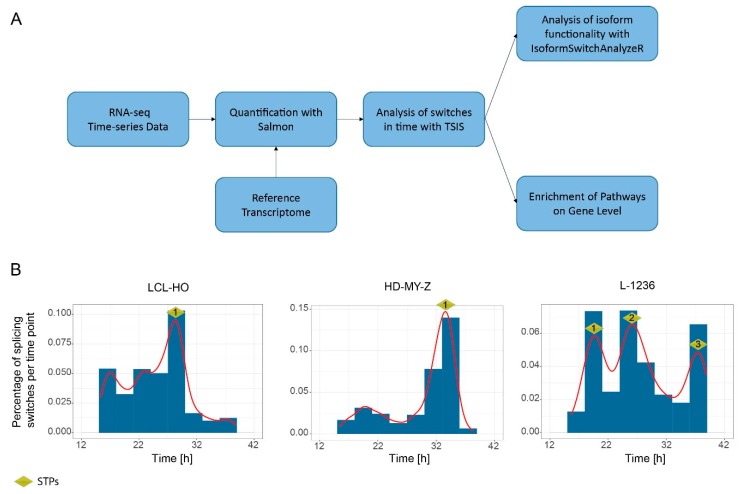
Analysis of isoform expression switching over time shows an enrichment of switching events in a time-specific manner (**A**) Schematic representation of the pipeline for the detection and analysis of isoform switching events. (**B**) Distribution of isoform-switching events for all genes over time. The LCL-HO cell line shows only one switch time peak (STP) between 27 h and 30 h ( STP1, 273 switching events), the HD-MY-Z cell line shows a STP between 33 h and 36 h (STP1, 783 switching events) and the L-1236 cell line shows three STPs between 18 h to 21 h (STP1, 166 switching events), 24 h to 27 h (STP2, 167 switching events) and 36 h to 39 h (STP3, 148 switching events).

**Figure 2 ijms-20-01182-f002:**
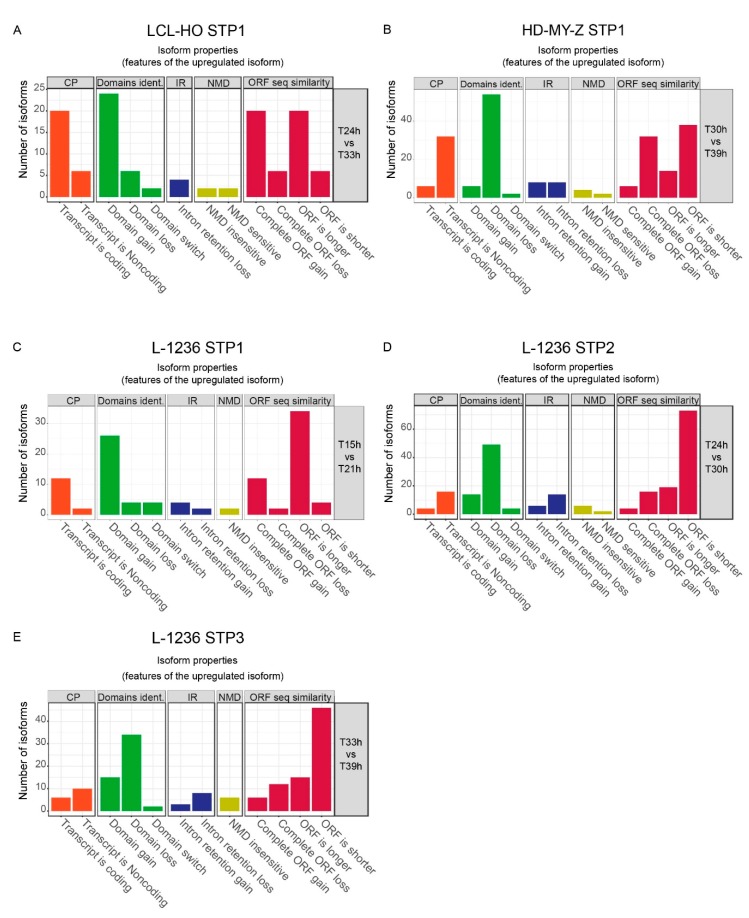
Computational analysis of the isoforms involved in the STPs shows that the loss of coding functionality and potential domain loss are the main consequences of changes in isoform expression. We performed a computational analysis only on the highly expressed genes and isoforms with changing expression in each STP. The analysis included coding potential (CP), identification of potential domains (domains ident.), possible intron retention (IR), nonsense-mediated decay (NMD) and open reading frame (ORF) length and availability (ORF seq similarity). (**A**) The STP between 27 h and 30 h for the LCL-HO cell line shows an increase in coding transcripts for the upregulated isoforms leading to gains in potential domains and increase of ORF length. (**B**) The STP between 33 h and 36 h for HD-MY-Z shows an increase of non-coding transcripts and a related loss of potential domains. The shorter ORF or its complete loss means that sequence elements coding for potential domains are lost. (**C**) The STP at 18 h to 21 h for the L-1236 cell line shows an increase in coding transcripts for the upregulated isoforms leading to gains in potential domains, and increase of ORF length. (**D**,**E**) The STPs at 24 h to 27 h and 36 h to 39 h (respectively) for the L-1236 cell line show an increase of non-coding transcripts and a related loss of potential domains with an increase in both complete ORF loss and shorter ORFs.

**Figure 3 ijms-20-01182-f003:**
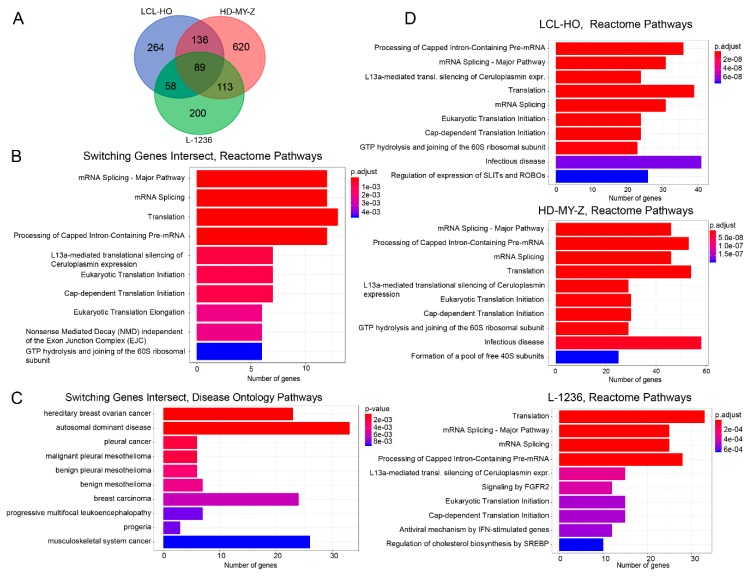
A core set of genes with switches between isoform expression over time is common to the cell lines analysed (**A**) We detected an intersection of 89 genes for the three cell lines, for the genes for which switching in isoform expression was detected in our analysis. (**B**) The top enriched pathways for the 89 genes are mRNA splicing and translation, as well as the processing of capped intron-containing pre-mRNA. (**C**) All but one of the top 10 enriched disease ontology terms associated with the 89 genes are different forms of cancer. (**D**) We analysed all genes for which switches in isoform expression over time were detected for the three cell lines. A set of common pathways including mRNA splicing, translation and ribosomal activity was identified for all cell lines, the L-1236 cell line shows an additional enrichment for the FGFR2 pathway.

**Figure 4 ijms-20-01182-f004:**
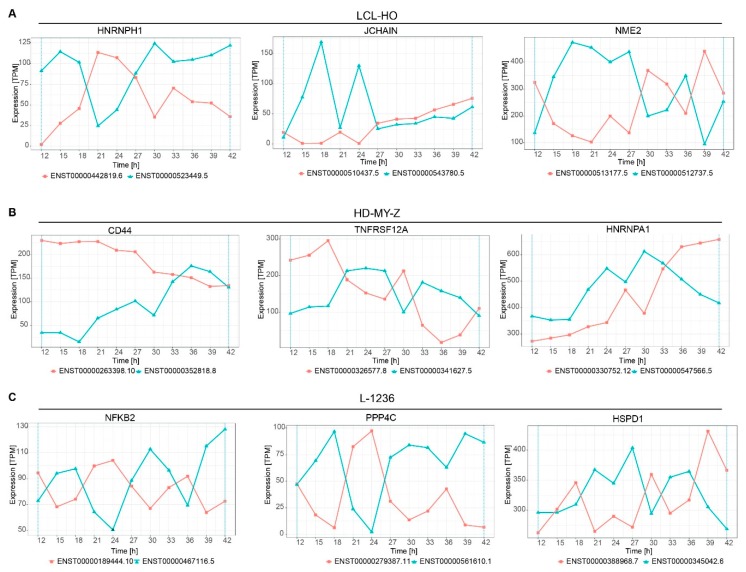
Several of the genes with largest expression difference between the switching isoforms are related to the tumour necrosis factor (TNF)-signalling pathway. (**A**) Out of the top 20 genes with largest expression difference between the isoforms for the LCL-HO cell line are the *JCHAIN*, *NME2* and *HNRNPH1* genes, which depict remarkable difference in expression patterns over time. (**B**) Central genes involved in cancer metastasis such as *CD44* undergo switching in isoform expression in HD-MY-Z. *TNFRSF12A* is related to the TNF-signalling pathway. (**C**) The changes in the expression of NFKB2 in L-1236 are associated to the TNF signalling pathway, and suggest possible rhythmicity of isoform expression switching. Also the expression changes between isoforms for *PPP4C* show a circadian pattern.

**Figure 5 ijms-20-01182-f005:**
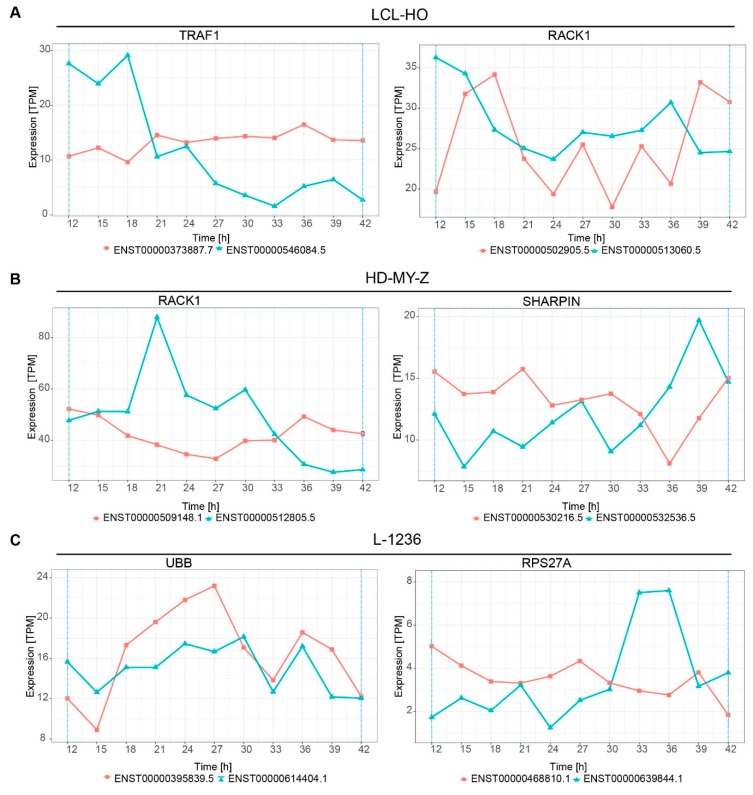
Multiple genes from the TNF pathway show switching between the isoform expression over time. (**A**) For the LCL-HO cell line the *TRAF1* and *RACK1* genes show a differential expression of the corresponding isoforms. (**B**) For the HD-MY-Z cell line, the *RACK1* gene shows switching behaviour between the isoforms. Multiple isoforms of the gene show a switch phenotype over time. The *SHARPIN* gene shows switching of isoform expression at 36h-39h, corresponding to STP1 for the cell line HD-MY-Z. (**C**) For the L-1236 only the genes *UBB* and *RPS27A* are affected by switching in time between isoform expression within the TNF pathway.

**Figure 6 ijms-20-01182-f006:**
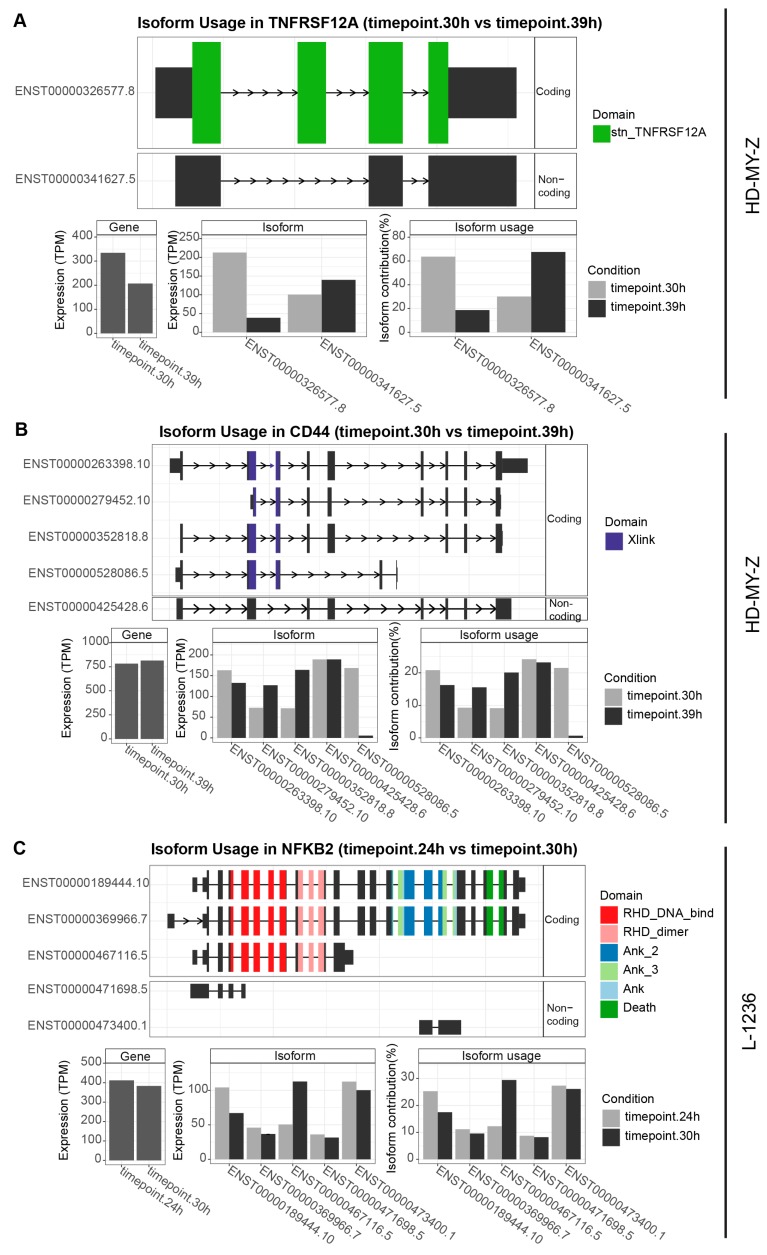
Genes related to the TNF signalling pathway show switching between isoform expression with potential functional implications. (**A**) Based on the analysis of switching behaviour between isoforms (Figure 4A), we further investigated the potential functionality of the isoforms. The two isoforms of *TNFRSF12A* have different predicted functionality. ENST00000326577.8 has predicted coding functionality and a stn_TNFRSF12A domain, whereas the ENST00000341627.5 is predicted to be non-coding. (**B**) The *CD44* gene with its multiple isoforms illustrates the importance of alternative splicing: the total gene expression is almost unchanged between the time points 30 h and 39 h, however the individual isoforms show a larger deviation in expression. Most notably, the ENST00000528086.5 isoform, predicted to be coding, is almost absent at 39 h. (**C**) In the L-1236 cell line we detected changes in isoform balance in the *NFKB2* gene (Figure 4C). The total expression of the gene shows only a minor change between the time points, however the isoform ENST00000467116.5 lacking the important death-domain is upregulated at 30 h as compared to 24 h (STP2).

**Figure 7 ijms-20-01182-f007:**
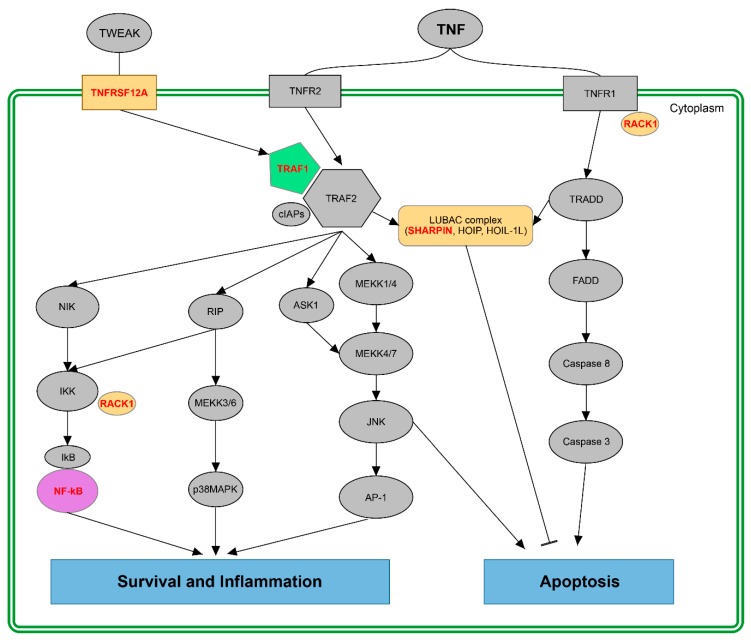
Elements within the TNF pathway possess a time-dependent switching behaviour in isoform expression: TNF can activate different pathways to induce cell survival, inflammation or apoptosis. Cell apoptosis is induced by the activation of Caspase 8 through TRADD and FADD. TRAF1/TRAF2 complex mediates cell survival and inflammation via the JNK-dependent cascade, MEKK cascade and the NF-kB activation by RIP. Elements, which display time-dependent switches in isoform expression are marked in colour with their associated genes marked red. Yellow: elements identified in the HD-MY-Z cell line. Green: elements identified in the LCL-HO cell line. Purple: elements identified in the L-1236 cell line.

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
