# Peer review of "Temporal Splicing Switches in Elements of the TNF-Pathway Identified by Computational Analysis of Transcriptome Data for Human Cell Lines"

_ijms, 2019, doi:10.3390/ijms20051182_

Reviewer 1 Report

The article entitled “Temporal splicing switches in elements……..human cell lines” by Genov et al. describes the importance of alternative splicing in the TNF pathway by using computational analysis of the transcriptome data for human cell lines.

The study is well planned and is supported by conclusive results in support of the role of temporal alternative splicing in isoform production in TNF pathway and impact on tumorigenesis.

I have the following minor suggestions for the manuscript:

1.       The spelling of alternative is wrong in line 21.

2.       Lines 54-57 have language error. Please rephrase or rectify.

3.       Lines 54-57 have language error. Please rephrase or rectify.

4.       Write full form of TNF at its first usage in the text.

5.       Line 515, Replace release with released.

Author Response

Dear Dr. Wu,

Thank you very much for conveying to us the reviewers’ comments on our manuscript ‘Temporal splicing switches in elements of the TNF-pathway identified by computational analysis of transcriptome data for human cell lines (ijms-456023) authored by Nikolai Genov, Alireza Basti, Mónica Abreu and Angela Relógio.

We are pleased to see that the reviewers were very positive about our work, considered its novelty, and found it deserves publication in IJMS. They further state that our paper is “well written and follows a logical path” that our results describe “the importance of alternative splicing in the TNF pathway “, and that our bioinformatics analyses is well planned and is supported by conclusive results”. The reviewers appreciated that our paper contributes with new findings to a better understanding of “the role of temporal alternative splicing in isoform production in TNF pathway and impact on tumorigenesis”.

Here, we submit a new version of our paper, in which we have addressed the minor changes suggested by the reviewers.

We include a revised version of the manuscript with tracked changes, as well as a clean copy. We are confident that we have satisfactorily addressed all issues raised by the reviewers and that our work, is now ready for publication in IJMS.

Looking forward to hearing from you soon.

With kind regards,

Angela Relógio

Detailed response to the reviewers’ comments

Reviewer #1:

The article entitled “Temporal splicing switches in elements……..human cell lines” by Genov et al. describes the importance of alternative splicing in the TNF pathway by using computational analysis of the transcriptome data for human cell lines.

The study is well planned and is supported by conclusive results in support of the role of temporal alternative splicing in isoform production in TNF pathway and impact on tumorigenesis.

We thank the reviewer for the very positive evaluation of our work and her/his awareness concerning the relevance of our data towards a better understanding of the role of circadian splicing in isoform production in TNF pathway.

R1.1.

I have the following minor suggestions for the manuscript:

The spelling of alternative is wrong in line 21.

Modified as requested.

Changes to the manuscript:

p. 2, Introduction

Alternative splicing (AS) belongs to one of the key biological processes, which regulates gene expression diversity and function in eukaryotes.

R1.2, R1.3.

Lines 54-57 have language error. Please rephrase or rectify.

Modified as requested.

Changes to the manuscript:

p. 2, Introduction

Additionally, several splicing factors (e.g. hnRNP E1, U2AF65 and NOVA-1/2) were reported to be differentially expressed in the HL cell lines as compared to the B lymphoblastoid cell line (LCL-HO). These findings point to the relevance of splicing in tumorigenesis for this cellular model of HL [13]. 

R1.4.

 Write full form of TNF at its first usage in the text.

Modified as requested.

Changes to the manuscript:

p. 2, Introduction

One of the pathways for which published data points to a role of AS in the regulation of key players of the signalling cascade is the TNF (Tumour Necrosis Factor) pathway

R1.5.     

Line 515, Replace release with released.

Modified as requested.

Changes to the manuscript:

p. 19, Data Access

The RNA-sequencing data was submitted to the GEO repository and will be released upon publication of the manuscript.

Reviewer #2:

Several studies have implicated the role of aberrant splicing in various human diseases. Almost every hallmark of cancer can be characterized by numerous examples of altered splicing associated with proto-oncogenes, tumor suppressor genes, or other genes of various signaling pathways.

In this article, the authors investigated the time-dependent production of splice variants in two stages of Hodgkin-Lymphoma cell lines and compared them to the lymphoblastoid cell line. Authors identified several genes of the TNF pathway to undergo a time-dependent variation in isoform production. The article is well written and follow a logical path. And authors correctly conclude that the significance and relevance of the temporal regulation of the TNF pathway in tumorigenesis need further investigation.

We thank the reviewer for her/his positive evaluation of our work.

Reviewer 2 Report

Several studies have implicated the role of aberrant splicing in various human diseases. Almost every hallmark of cancer can be characterized by numerous examples of altered splicing associated with proto-oncogenes, tumor suppressor genes, or other genes of various signaling pathways.

In this article, the authors investigated the time-dependent production of splice variants in two stages of Hodgkin-Lymphoma cell lines and compared them to the lymphoblastoid cell line. Authors identified several genes of the TNF pathway to undergo a time-dependent variation in isoform production. The article is well written and follow a logical path. And authors correctly conclude that the significance and relevance of the temporal regulation of the TNF pathway in tumorigenesis need further investigation. 

Author Response

(The authors gave the same response as above.)
